# Nitrogen-Dioxide Remains a Valid Air Quality Indicator

**DOI:** 10.3390/ijerph17103733

**Published:** 2020-05-25

**Authors:** Hanns Moshammer, Michael Poteser, Michael Kundi, Kathrin Lemmerer, Lisbeth Weitensfelder, Peter Wallner, Hans-Peter Hutter

**Affiliations:** 1Department of Environmental Health, Center for Public Health, Medical University Vienna, 1090 Vienna, Austria; michael.poteser@meduniwien.ac.at (M.P.); michael.kundi@meduniwien.ac.at (M.K.); kathrin.lemmerer@meduniwien.ac.at (K.L.); peter.wallner@meduniwien.ac.at (P.W.); lisbeth.weitensfelder@meduniwien.ac.at (L.W.); hans-peter.Hutter@meduniwien.ac.at (H.-P.H.); 2Department of Hygiene, Medical University of Karakalpakstan, Uzbekistan, Nukus 230100, Uzbekistan

**Keywords:** nitrogen dioxide, indicator, diesel-powered vehicles, dieselgate, proxy, vienna

## Abstract

In epidemiological studies, both spatial and temporal variations in nitrogen dioxide (NO_2_) are a robust predictor of health risks. Compared to particulate matter, the experimental evidence for harmful effects at typical ambient concentrations is less extensive and not as clear for NO_2_. In the wake of the “Diesel emission scandal—Dieselgate”, the scientific basis of current limit values for ambient NO_2_ concentrations was attacked by industry lobbyists. It was argued that associations between NO_2_ levels and medical endpoints were not causal, as NO_2_ in older studies served as a proxy for aggressive particulate matter from incineration processes. With the introduction of particle filters in diesel cars, NO_2_ would have lost its meaning as a health indicator. Austria has a high percentage of diesel-powered cars (56%). If, indeed, associations between NO_2_ concentrations and health risks in previous studies were only due to older engines without a particle filter, we should expect a reduction in effect estimates over time as an increasing number of diesel cars on the roads were outfitted with particle filters. In previous time series studies from Vienna over shorter time intervals, we have demonstrated distributed lag effects over days up to two weeks and previous day effects of NO_2_ on total mortality. In a simplified model, we now assess the effect estimates for moving 5-year periods from the beginning of NO_2_ monitoring in Vienna (1987) until the year 2018 of same and previous day NO_2_ on total daily mortality. Contrary to industry claims of a spurious, no longer valid indicator function of NO_2_, effect estimates remained fairly stable, indicating an increase in total mortality of previous day NO_2_ by 0.52% (95% CI: 0.35–0.7%) per 10 µg/m^3^ change in NO_2_ concentration.

## 1. Introduction

The “fraud of Volkswagen” [1] has precipitated a discussion on the health impact of compression–ignition engine emissions from vehicles [2]. Holland et al. concluded that estimated financial- and health-related damages from excess emissions would greatly exceed possible benefits due to economic returns from marketing. The assumptions of Holland et al. were strongly criticized by others [3]. While this discussion was still on-going, frequent exceedances of European air quality limit values of nitrogen dioxide (NO_2_) have triggered restrictions in many cities, for example, in Germany, for older diesel passenger cars. Again, these measures were deemed “highly debatable” [4]. 

The question about the health effects attributable to current concentrations of NO_2_ in ambient air is still controversially debated among scientists. The answers of the WHO working group [5] illustrate that point concisely: 

“Elevated health risks associated with living in close proximity to roads is unlikely to be explained by PM_2.5_ mass since this is only slightly elevated near roads. In contrast, levels of such pollutants as ultrafine particles, carbon monoxide, NO_2_, black carbon, polycyclic aromatic hydrocarbons, and some metals are more elevated near roads. Individually or in combination, these are likely to be responsible for the observed adverse effects on health. Current available evidence does not allow discernment of the pollutants or pollutant combinations that are related to different health outcomes, although association with tailpipe primary PM is identified increasingly.”

“Many studies, not previously considered, or published since 2004, have documented associations between day-to-day variations in NO_2_ concentration and variations in mortality, hospital admissions, and respiratory symptoms. Also, more studies have now been published, showing associations between long-term exposure to NO_2_ and mortality and morbidity. Both short- and long-term studies have found these associations with adverse effects at concentrations that were at or below the current EU limit values, which for NO_2_ are equivalent to the values from the 2005 global update of the WHO air quality guidelines. Chamber and toxicological evidence provides some mechanistic support for a causal interpretation of the respiratory effects. Hence, the results of these new studies provide support for updating the 2005 global update of the WHO air quality guidelines (WHO Regional Office for Europe, 2006) for NO_2_, to give: (a) an epidemiologically based short-term guideline value; and (b) an annual average guideline value based on the newly accumulated evidence. In both instances, this could result in lower guideline values.”

In short, the general scientific consensus appears to be that NO_2_ is a valuable proxy of high-resolution spatial distribution of poor air quality. If it is, by itself, causing adverse health effects at current ambient concentrations remains debatable. Using cross-sectional data from Linz, Austria, we have previously shown that children living in households cooking with natural gas have poorer lung function than their peers from households with electrical stoves [6] and could later also demonstrate the same in a pooled study from Europe and North America [7]. We originally argued that these findings were likely due to NO_2_ because it is the most prominent respiratory toxicant produced by the gas flame. However, in our pooled study, discussion with co-authors resulted in a more cautious interpretation because gas flames, although normally producing only a very low amount of particulate matter mass, could still produce a high number of ultra-fine particles. Similar to NO_2_ from road traffic exhaust, indoor levels of NO_2_ could signify complex pollutant mixtures.

This scientific discussion about NO_2_ being an indicator of effects of a pollutant mixture or itself the causal factor was taken up by industry lobbyists and representatives of car owners, their arguments precipitating a pointed debate in the public media [8,9], thereby gaining strong political momentum. In this debate, not only the public health relevance of NO_2_ was questioned, but limit values for ambient air quality in general.

Austria has a high percentage of more than 50% of diesel-powered passenger cars [10]. We have repeatedly shown that NO_2_ represents a valuable marker of acute health effects in time-series studies from Vienna [11,12] and from other Austrian cities [13]. In these previous time-series studies from Vienna, we have examined particulate air pollution and NO_2_ for the years 2000–2004 [11], and ozone and NO_2_ for the years 1991–2009 [12]. In the first study, we demonstrated that a distributed lag model provides more robust and higher risk estimates than a single lag model. In the second study, we compared previous and same-day NO_2_ and found stronger effects on previous day NO_2_ (lag 1) that were not affected by controlling for ozone.

If industry claims are correct that, along with improved particle filter technology, ambient levels of NO_2_ have lost their indicator function for adverse health effects of diesel exhaust emissions, we would expect decreasing risk estimates for NO_2_ in time series studies, especially in Austria with a large proportion of diesel cars. To test this hypothesis, we investigated a longer time-span of data for Vienna, covering the beginning of NO_2,_ monitoring from 1987 till the end of 2018.

## 2. Materials and Methods 

### 2.1. Daily Air Pollution Data

Daily mean values of NO_2_ from all monitoring stations in Vienna were downloaded from the European Environment Agency (EEA) database [14]. A correlation matrix of all stations was built and a station with a long documented observation period and high correlation with the other stations was chosen as representative for all of Vienna. Data gaps from records of that station were filled with estimated values from linear regression based on data from another available station that was selected by maximal correlation. NO_2_ monitoring in Vienna started in 1987. As in the first year of monitoring, only one station was operative; interpolation of missing values was not possible for this year.

### 2.2. Meteorological Data

Meteorological data (daily mean temperature, daily mean relative humidity and daily mean air pressure) were abstracted from the annual reports of the Austrian Meteorological Service [15]. The monitoring station is placed at “Hohe Warte” in the Western hilly part of Vienna, outside of the central heat island area.

### 2.3. Daily Mortality Counts

Mortality data have been obtained from the national Austrian Statistics Institute (Statistik Austria). More details about the data structure are reported by Weitensfelder and Moshammer [16]. For each death occurring in Austria since 1 January 1970, the following information was provided: age (in years), sex, date of death, most recent place of residence (district), and primary cause of death. The latter information was provided as International Code of Diagnoses (ICD) version 8 (ICD8) until 1979, as ICD9 until 2001, and as ICD10 from 2002 onwards. After 2015, cause of death was no longer available due to data protection concerns. Because of the changes in diagnostic coding and the lack of cause-of-death data for the last 4 years, only total daily mortality was considered, selecting all residents of any of the 23 districts of the city of Vienna.

### 2.4. Whole Period General Additive Model (GAM)

Covering the whole observation period (1987–2018) a General Additive Model (GAM) was constructed. For the sake of comparability, the same parameters were used as in Neuberger et al. [11]. In that study, degrees of freedom for the seasonal and temporal trends spline were selected to minimise partial autocorrelation. The best fitting lag for temperature and humidity, and the respective changes between consecutive days, were selected step by step based on the Akaike Information Criterion (AIC) [17]. In that previous paper, the best fit was with 6 degrees of freedom per year, lag 0 for temperature, lag 1 for difference in temperature to the previous day, and lag 2 for relative humidity and for the difference in humidity to the previous day. 

We finally ran three general additive models, one without air pollution data to calculate the residuals, one to estimate linear effects of same day and one of previous day NO_2_ concentrations on daily all-cause mortality.

### 2.5. Simplified Parametric Poisson Regression

For ease of calculation in repeated models, a simplified regression model was built, replacing splines by parametric factors. In the GAM, long-term trend and seasonal variation in daily mortality and temperature effects were the most influential non-linear factors. They were replaced on the one hand by a linear long-term time trend and by an annual sine-cosine function, and on the other hand by a quadratic polynomial for temperature. We have already reported that this Poisson model provides a good regression fit with little evidence of over-dispersion [18]. To this model, we either added same- or previous-day NO_2_. While the effect estimates in the linear regression on residuals (see below) directly represent excess daily deaths per 1µg/m^3^ increase in NO_2_, the effect estimates of the Poisson regression are the natural logs of relative risks. The relative risk minus 1 multiplied by the average number of daily deaths represents excess daily deaths. For ease of interpretation, we calculate relative risks (RR) and display estimates (and 95% confidence intervals) as percentage changes per 10 µg/m^3^. We calculated the Poisson coefficient per 10 µg/m^3^ by multiplying the original coefficient with 10. Next, we transformed the coefficient into a relative risk exp(ß*10), subtracted 1 and multiplied by 100 to display estimates as percentages: (exp(ß*10) − 1)*100.

### 2.6. Moving 5-Year Periods

The parametric Poisson regression was then also run for multiple overlapping 5-year periods (1987–1991, 1988–1992, and so on, until 2014–2018). Effect-estimates for same- and previous-day NO_2_ were graphically explored, depicting point estimate and the 95th confidence interval.

### 2.7. Annual Regression Models of NO_2_ Concentrations on Residuals

Because the 5-year periods are overlapping, periodic effect estimates are not independent of each other. A regression analysis of temporal trends, therefore, would generate inflated precision estimates. On the other hand, estimates based on single years are unstable because seasonal variation can only be estimated when more years are observed, and examining only non-overlapping periods reduces the number of periods.

Therefore, we also examined an alternative model: effects of season and long-term trend as well as meteorological parameters were estimated, utilizing a GAM for the whole observation period (1970–2018). The residuals of that GAM were supposed to be normally distributed. The effect of same- and previous-day NO_2_ was examined by linear regression on the residuals for each year separately. 

All calculations were done with STATA 15.1 (Stata Corp: College Station 2017, TX, USA) [19].

## 3. Results

Monitoring of NO_2_ began in Vienna in the year 1987. The first monitoring station was situated at Lobau (LOB), which is a floodplain forest along the Danube in the East of Vienna. This monitoring station was clearly dedicated for eco-toxicological monitoring and not for human-public-health-related data. Only in the following year, 1988, were more stations installed in the urban area. 

Over the years, in total, 18 stations reported NO_2_ concentrations. Upon inspection of iterative pairwise correlation, five stations were selected that were most strongly correlated with each other: Belgradplatz (BELG), Stefansplatz (STEF), Taborstrasse (TAB), Gaudenzdorf (GAU) and AKH (Table 1). A description of the monitoring sites (in German) can be found at https://www.wien.gv.at/umwelt/luft/messstellen/index.html.

BELG is an urban background station in the south-east of Vienna, STEF is in the city center in the pedestrian zone, TAB is a rather central curb-side station, GAU is an urban background station in the central western part of Vienna and AKH is the acronym for the General Hospital (“Allgemeines Krankenhaus” in German) west of the center, in the hospital gardens, but not far from a busy road (“Währinger Gürtel”). We have utilized data from AKH in previous studies [20] and found it representative of a wider area. Unfortunately, this station was moved within the area of the hospital in 1996/1997 causing a longer interruption of the timeline. Therefore, in this study we used data from BELG, replaced by data from (in that order) STEF, TAB, GAU and AKH in the rare case of missing data. Only for the first year (1987), did we make use of data from LOB.

The correlation coefficient per year between BELG and each of the other four stations increased over the years and remained fairly stable after the year 2000 (Figure 1).

NO_2_ measured at or estimated for BELG had an average concentration of 37.1 µg/m^3^ with a median of 35.0 µg/m^3^. Over the years (Figure 2), it declined significantly by 0.5 µg/m^3^ per year (*p* < 0.001). Concentrations peaked each winter.

As described in more detail in another study by our group [16], daily mortality declined over time until after the year 2005 and remained stable thereafter (Figure 2). During the whole observation period, beginning in 1970, an average number of about 56 daily deaths were recorded (with higher counts in winter than in summer). In the years with NO_2_ measurements (starting in 1987), the daily average number of deaths in Vienna was about 49. In recent years, besides a commonly observed peak of daily deaths in winter, a summer peak became increasingly obvious. Nevertheless, the sine-cosine-based model still reflected the winter peak, and the summer peak could be explained by the temperature effect that resembled a U-shaped function and was well represented by a quadratic polynomial of same-day temperature.

For the whole period (1987–2018), effect estimates both for same- and previous-day NO_2_ were stronger in the GAM than in the parametric Poisson model. While in the GAM, NO_2_ from both days had a significant effect on total mortality in the Poisson model, only previous-day NO_2_ effects remained significant (Table 2).

The overlapping five-year intervals only rarely showed significant effects of single-day NO_2_. Effect estimates varied over the course of time without any indication of declining estimates or even with a hint of increasing effect strengths with previous day NO_2_ (Figure 3).

The effect estimates of NO_2_ on the residuals were, for most years, above the null (Figure 4) and showed a significant increase over the years (weighted by the standard errors): *p* for same day NO_2_: 0.007, for previous day NO_2_: 0.005. 

The effect estimates of previous day NO_2_ examined in the GAM in a spline model with 2 degrees of freedom indicated a significant deviation from linearity (*p* = 0.0003) (Figure 5).

The residuals of the GAM displayed a fairly normal distribution (Skewness of 0.3 and Kurtosis of 3.5). The variance of 67 is in accordance with the assumption of a Poisson distribution of the original data. 

## 4. Discussion

Regarding the observation of increasing effect estimates per a fixed change in NO_2_ concentration, while the concentrations, on average, are declining, we can think of four possible explanations: (1)In the early years, when concentrations were still higher, there might have been a larger variation between stations, rendering the one station BELG less representative of the city-wide exposure. This we checked by examining the annual correlation coefficients between BELG and selected stations in other regions of Vienna. Indeed, the correlation coefficients between BELG and the four other closely correlated stations (STEF, TAB, GAUD, and AKH) increased over time. However, the increase was most prominent in the first years and leveled-off after the millennium (Figure 1). This cannot explain the increase in the effect estimates in the last decade;(2)The observation might simply be a result of a non-linear relationship between concentration and risk with a kind of saturation effect at higher concentrations. Indeed, a spline function for NO_2_ in the GAM does display a convex form with a very broad confidence interval because of a scarcity of observations and no further substantial increase at higher concentrations, but a clear increase at concentrations below about 60 µg/m^3^ (Figure 5). Instead of concentration–response function being the cause of the observed increase in effect estimates over time, it could just reflect the effect that, in the earlier years with higher ambient concentrations, the effects were also weaker;(3)If NO_2_ were causal for the observed increase in deaths and not just a proxy for a more complex pollution mixture, lower concentrations would lead to the longer survival of those people that are more vulnerable to NO_2_. If, then, in spite of lower concentrations in general, a now-rarer concentration peak occurred, more people would be affected, thus increasing the overall effect estimates;(4)If, on the other hand, NO_2_ were just a proxy of a more complex pollution mixture, and if the major source of that pollution mixture had undergone some technological changes reducing NO_X_ emissions, but not the emissions of the other harmful substances, then increasingly lower NO_2_ concentrations would serve as a proxy of the same amount of pollutants. However, industry lobbyists are claiming the opposite: diesel engines have become substantially cleaner with the new emission standards, with only NO_X_ emissions remaining high or even increasing.

Whatever the reasons, we do observe an increase in the effect-estimates and not a decrease. Overall, effect-estimates from the GAM were comparable to effect-estimates proposed by the HRAPIE project [21], and also to our previous findings [11,12]. The overall effect-estimates from the parametric Poisson model were smaller. That model has demonstrated good performance for investigating long-term trends in temperature effects [16] and even for investigating the impact of daylight-saving schemes [18]. However, that model was not precise enough for an unbiased estimate of the more subtle effects of air pollution. Nevertheless, we are confident that any biases would have affected all 5-year groups in a similar way. As we were not interested in the exact magnitude of the effects or even the true cumulative effects over multiple lags, but rather in the relative change in these effects over time, we are confident that the less computationally challenging model provided valid information in that regard.

Both GAM and parametric Poisson model depend on data from multiple years to ensure reliable control of seasonal variation. This stands in conflict with the aim of analyzing long-term trends on a year-by-year basis. To overcome this problem, we also investigated an alternative approach: we modeled the effects of long-term trend, seasons, and meteorology in a standard GAM and estimated NO_2_ effects on an annual basis on the residuals of the GAM. This approach is unorthodox in that sense, as it assumes constant absolute effects per a given change in NO_2_ concentration. Usually, effects relative to the total number of cases are assumed. Indeed, it is not plausible that a change in NO_2_ concentration per 10 µg/m^3^ will always produce the same number of extra deaths, no matter how large the overall average mortality rate. However, if effect-estimates are constant and absolute numbers of average total deaths are declining, we would expect declining absolute effect-estimates. Increasing absolute effect-estimates in spite of declining overall numbers underlines the message of increasing relative effect-estimates. Maybe, with this consideration in mind, it is not so surprising that the strongest increase in absolute effect-estimates (Figure 4) is visible in the years after the millennium when average total mortality remained stable (Figure 2). 

An observational study cannot replace experimental evidence and will not be able to solve the question of whether NO_2_ by itself causes the adverse effects or if it rather serves as an indicator of a more complex pollution mixture. However, one thing is certain: NO_2_ still remains a valid indicator of air quality that is relevant to and predictive for human health [22,23]. This might be especially true for susceptible groups like, e.g., asthmatics [24,25].

Apart from the issue of the direct health effects (of NO_2_ or of the mixture it signifies), nitrogen oxide also plays an important role in atmospheric chemistry [26,27] and in ecosystems [28,29]. Therefore, there are many valid reasons for reducing NO_X_ emissions that must be kept in mind.

## 5. Conclusions

Modern compression–ignition engines produce higher amounts of nitrogen oxide per amount of diesel consumed. This is mostly due to higher ignition temperatures that also increase fuel efficiency. However, technologies to reduce NO_X_ emissions have already been developed and are already in use in heavy duty vehicles. The car manufacturers tried to avoid the additional costs of these technologies for their passenger cars, claiming that nitrogen dioxide from modern diesel-powered vehicles is no longer dangerous for health and that existing limit values should be reconsidered. 

There are many good reasons for keeping tight NO_2_ limit values. This study adds to the growing evidence base that supports strict limit values from an epidemiological point of view.

Strict emission control also, and especially regarding urban road traffic, is key to ensuring good urban air quality [30,31,32].

## Figures and Tables

**Figure 1 ijerph-17-03733-f001:**
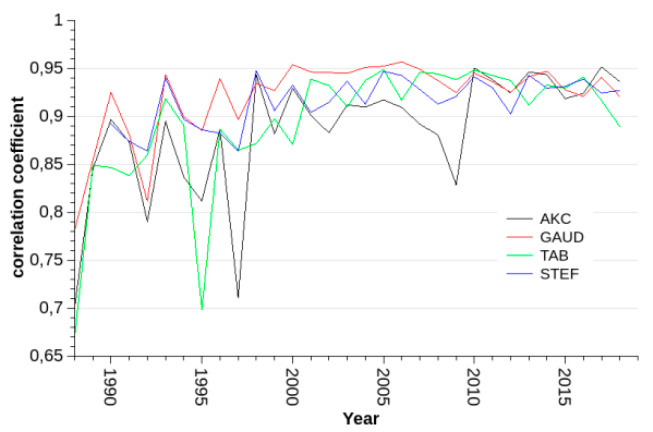
Correlation coefficients between the stations and Belgradplatz (BELG) per year.

**Figure 2 ijerph-17-03733-f002:**
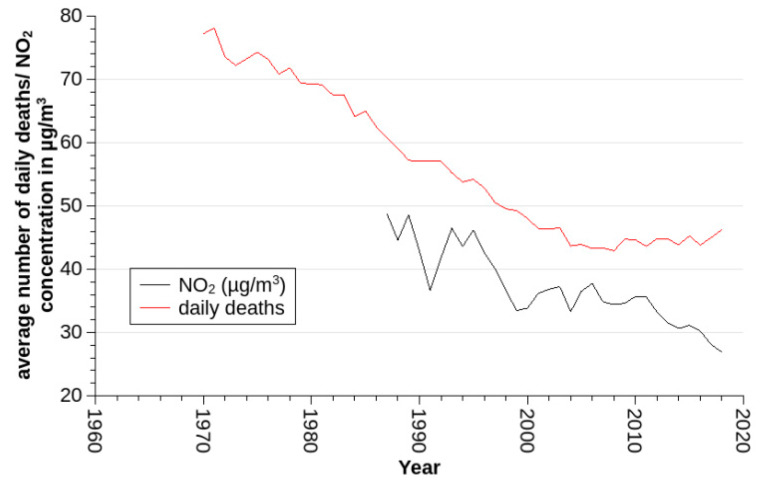
Average daily number of deaths (number) and concentration of NO_2_ (µg/m^3^) per year.

**Figure 3 ijerph-17-03733-f003:**
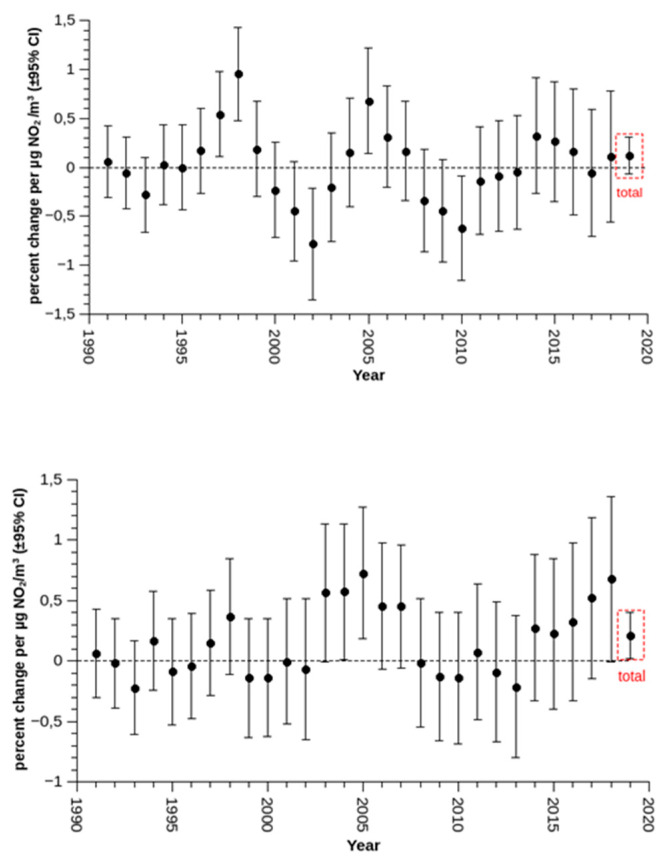
Top: Effect estimates (percent change of daily mortality per 10 µg/m^3^ increase in NO_2_) of same day NO_2_ (5 year sliding estimates until, Poisson model). Bottom: Effect estimates (percent change in daily mortality per 10 µg/m^3^ increase in NO_2_) of previous day NO_2_ (5-year sliding estimates until, Poisson model).

**Figure 4 ijerph-17-03733-f004:**
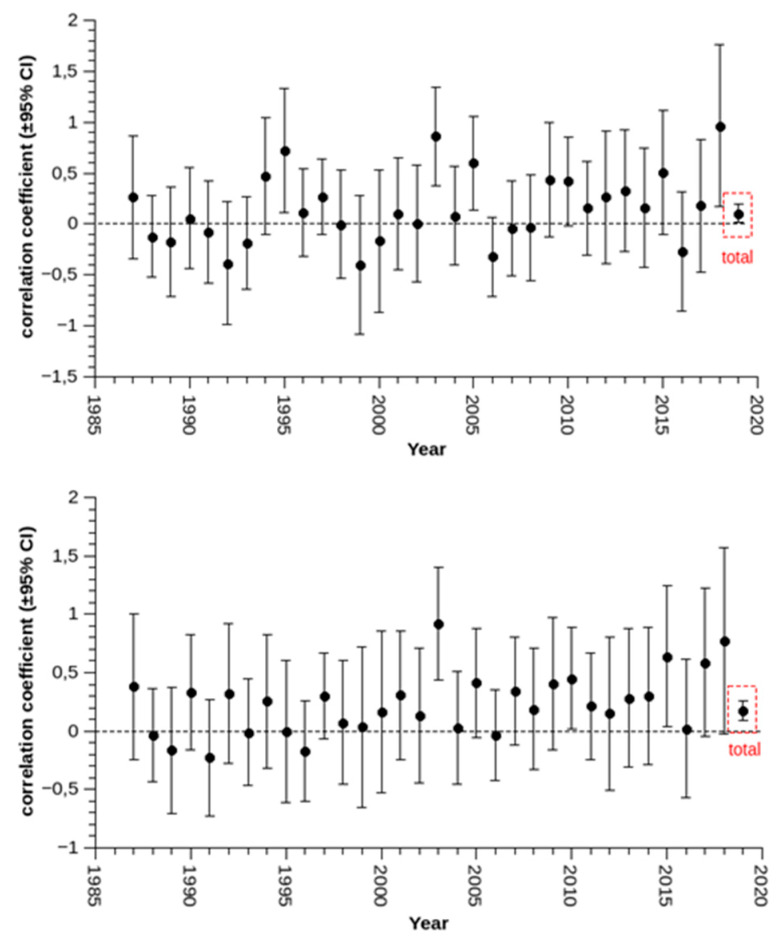
Top: Effect estimates of same day NO_2_ on residuals of GAM. Bottom: Effect estimates of previous day NO_2_ on residuals of GAM.

**Figure 5 ijerph-17-03733-f005:**
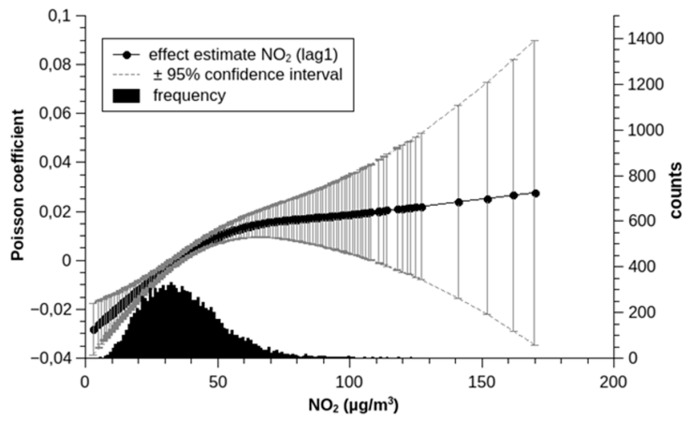
Concentration-response-function of previous day NO_2_ in a spline model (2 degrees of freedom).

**Table 1 ijerph-17-03733-t001:** Air quality stations and correlation for daily NO_2_ with Belgradplatz (BELG).

Station	Operated From	Operated Till	Pearson’s R
BELG	19.03.1988	31.12.2018	-
STEF	01.01.1990	31.12.2018	0.8871
TAB	01.01.1988	31.12.2018	0.8715
GAUD	16.03.1988	31.12.2018	0.8605
AKH	01.01.1988	31.12.2018 ^1^	0.8604
LOB	01.01.1987	31.12.2018 ^2^	0.5127
LIES			0.8652
ZA			0.8263
KE			0.7848
LAA			0.7844
STAD			0.7646
MBA			0.7563
FLO			0.7501
SCHA			0.7144
HER			0.6735
RINN			0.6458
KEND			0.6255
JAEG			0.4621

^1^ With a gap in 1997; ^2^ Incomplete data in 1987.

**Table 2 ijerph-17-03733-t002:** Effect estimates (% change per 10 µg/m^3^ NO_2_).

Model	Same Day NO_2_			Previous Day NO_2_		
	Estimate	95% Confidence Interval		Estimate	95% Confidence Interval	
GAM	0.33	0.15	0.51	0.52	0.35	0.70
Poisson	0.12	−0.07	0.30	0.21	0.02	0.40

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
