# Peer review of "Nitrogen-Dioxide Remains a Valid Air Quality Indicator"

_ijerph, 2020, doi:10.3390/ijerph17103733_

Round 1

Reviewer 1 Report

Many thanks for submitting the paper for possible consideration in the journal. 

Reading through the work, there are areas in the paper that will benefit from improvement to help bring the quality to a desired level. 

i. line 170-190 should be moved to the material and method section possibly create a subtitle "description of study area" as this is what is been considered herein than result in my opinion. 

ii.  It was not clear the rationale why figure 4 and 5 were presented in the discussion section as against result (considering that both are results from the primary study been presented). Consider reverting same to the result section 

iii. the discussion section would require further tidying in the light to the recommeded action in (ii) above. 

Iv. Overall, a quick run through the entire paper will as well help improve its overall quality  

Reviewer 2 Report

The work is interesting, I have no comments on the methodological part, text clarity, maybe I am less convinced by the interpretation of the results, e.g. the risk of mortality due to NO2 concentrations is higher with decreasing concentrations. However, I agree with the final conclusion that despite these doubts, NO2 is an important indicator of air quality and a threat to human health.

My comment is, that  correct interpretation depends on the data  nature and how it is received. Here I have no doubt about the correctness of the analyzes. But "Dry" numbers were obtained. Remember that statistical tests do not prove the hypothesis is true or false. It only speaks of the probability of the hypothesis being true and I believe that the authors should stress this point and introduce some doubts into the results obtained.

  1. 131 “In the GAM, long-term and seasonal variation and temperature were....- it is not known what parameter “variation” applies to.

Round 2

Reviewer 1 Report

Many thanks for responding to the observation earlier raised. I have gone through each and satisfied with each. 

Best regards